# Clothes Make the Man—What Impact Does the Dress of Interprofessional Teams Have on Patients?

**DOI:** 10.3390/healthcare10102109

**Published:** 2022-10-21

**Authors:** Hans Joachim Roehrens, Jan P. Ehlers, Michaela Zupanic

**Affiliations:** 1Didactics and Educational Research in Health Care, Department of Medicine, Faculty of Health, Witten/Herdecke University, 58448 Witten, Germany; 2Interprofessional and Collaborative Didactics, Department of Medicine, Faculty of Health, Witten/Herdecke University, 58448 Witten, Germany

**Keywords:** clothing interprofessional teams, German family medicine team, doctors’ attire, patient groups, core values, team attire, FFP2 mask, coronavirus pandemic

## Abstract

Physicians’ attire seems to play an important role in the success of patient treatment. The classic doctor’s white coat initiates a strong signal to the patient and can have a determining effect on a successful doctor–patient relationship. In a quantitative online questionnaire study comprising 52 questions, participants were shown four photos of an interprofessional German family medicine team in varying attire. One important study feature relating to the ongoing coronavirus pandemic was that the team was portrayed wearing FFP2 masks in one photo. We measured core values regarding the team’s perception in terms of sympathy, competence, trust, choosing the practice as a personal health care provider, and wanting to participate in the team. The questionnaire was posted online between March and May 2021. It was accessed 1435 times and 906 sheets were qualified for statistical analysis. For the first time in this field of research, a practice team’s attire was investigated. We found a significant influence of different clothing on the perception of sympathy, competence, trust, elective practice, and team participation. Wearing an FFP2 mask promotes feelings of security and competence. The study shows that in times of fast social changes due to rapid digitalization and an ongoing pandemic, we should present ourselves in different ways as a medical team depending on the patient groups we are targeting and the feelings we want to evoke.

## 1. Introduction

Since time immemorial, the physician’s attire seems to be an important factor in the success of treatment. The classic white doctor’s coat initiates a strong signal to the patient that generates strong emotions and can have a determining effect on a successful doctor–patient relationship [1,2,3,4].

There has been a steadily increasing body of research in this field examining healthcare professionals’ attire and its impact on patients beginning in the 1990s [5,6,7]. By far the largest number of studies come from the Anglo-American region, with a setting located in the hospital setting [8,9,10]. Other interesting studies regarding physician attire come from the Middle East [11,12] and in a “primary care” setting from Japan [13,14,15]. Most of the studies from outside of Europe found the classic white coat to be the ideal doctor’s companion.

Data from European research on this highly exciting topic is limited and sparse. A Swiss questionnaire study from 2019 found that approximately 900 outpatients at a large Swiss university preferred their physicians to be dressed in white scrubs with a white coat. Interestingly the respondents rated the attire vastly differently regarding the type of physician, clinical setting, and demographic background [16]. A Belgian questionnaire study on GP attire which tested for patients’ confidence and comfort levels showed that the GPs attire had a definite influence on the tested values and that in general patients still mainly preferred a doctor in a professional outfit [17].

Although a great body of research is available referring to the various topics of single physician attire, none of the previous studies conducted internationally showed an interprofessional approach portraying a practice team in varying attire. Therefore, we designed our questionnaire to gather as much data as possible, targeting a multitude of potential evaluations by later analyzing the subgroups. We took a high data output into account and decided to investigate the subgroups subsequently. In addition, none of the existing studies worldwide had ever tested a practice team’s attire in combination with wearing an FFP2 mask as a consequence of the ongoing worldwide coronavirus pandemic. Only one study known to us tested for physicians’ personal protective equipment during the COVID-19 pandemic. Here, respondents approved of masks but less accepted eye and hand protection [18].

For our study, we chose an interprofessional approach because we think that nowadays, more than ever, the success of a treatment is not only guaranteed by a single physician, but by the well-coordinated cooperation of interdisciplinary teams. A 2012 study found the interprofessional care of patients to be at the center of contemporary, efficient, and patient-oriented medicine [19,20]. The setting of a modern German family practice working in a strictly outpatient setting with its multi-layered treatment fields and the care of a multitude of acute and chronic diseases offers the ideal prerequisite to investigate for the first time how an interprofessional team of German healthcare professionals (physicians, students, diabetes specialists, relieving care assistants, a specialist for outpatient medical care, and trainees) is perceived by patients. For our study, it is important to understand that German doctors and their teams have a less formal dress code as compared to other countries, especially those of British descent. Business suits, ties, slacks or shirts are not worn at all.

The focus of our study was the question of whether the three core values of the treatment process sympathy, competence, and trust were perceived. We also wanted to investigate, if there were differences in choosing the practice as a family practice or a potential workplace depending on different team clothing. We were eager to find out what effect a team wearing FFP2 masks had on the outcome in terms of perceiving core values. This fact had never been looked at before. We expected to gain new insights into the dynamics of team attire and its perception by different subgroups, as there had been no relevant research in this field before. As mentioned above, we collected a lot of data through our questionnaire study. In this paper, we focused on the five items of sympathy, competence, trust, choosing a practice, and teamwork as dependent variables with age and gender being subgroups. Further research will need to be undertaken to understand the impact of education and employment.

## 2. Methods

We designed a quantitative online questionnaire study conducted as part of a dissertation at the Chair of Didactics and Educational Research in Health Care at the University of Witten/Herdecke. The study received a positive vote from the ethics committee of the University of Witten/Herdecke (S-304/2020). The limitations (survey fatigue, response bias, inability to connect with people from remote areas/no internet) and advantages (quick data analysis, respondent anonymity, maximum reach with minimal effort cost cutting) of the survey method had been discussed before [21,22].

The online questionnaire, with a total of 52 questions, was created in German and English and made available for distribution via Limesurvey (limesurvey.uni-wh.de, accessed on 12 May 2021); see Appendix A. Everyone over the age of 18 was eligible to participate, as we all have a history of experiences and feelings as patients. The data protection declaration was made according to Article 13 DSGVO and consent for the questionnaire was obtained. Sociodemographic data (age, gender, place of residence, educational qualification, and professional field of activity) were requested in a “pull-down” menu. Two questions (answer options in percent from 1–100) were asked about the current COVID-19 pandemic. Participants were shown four photographs of an interprofessional family medicine team in different attire (see Figure 1).

The participants indicated on a sliding scale of 0–100 with the extreme values “not at all” or “very much” how sympathetic, competent, and trustworthy they judged the team shown. Furthermore, they were asked whether they would choose the team as a family practice and whether they could imagine working in the respective team. The question about choosing the practice as a family practice was supplemented by the following options (multiple choice): Routine, mild illness, severe illness, chronic illness, mental illness, and not at all. We further asked respondents to assign the team members shown in photos A and B to their supposed professions (physician, medical student, medical assistant, diabetes assistant, relieving care assistant, trainee medical assistant). It is important to stress, that there were no hints available relating to the respective team members’ professions. The depicted persons wearing white coats were only described as team members, not identified as doctors.

The questionnaire was posted online between March and May 2021, and the link was sent via mail distribution lists, Facebook, Instagram, LinkedIn, Xing, and healthcare provider/health insurance websites for distribution. It was accessed 1435 times and completed 935 times. After applying the inclusion criteria (place of residence: Germany, Austria, Switzerland, gender indicated), 906 sheets qualified for statistical analysis. The data were analyzed using descriptive statistics (means, standard deviations, minimum, maximum). Group comparisons were performed for categorical variables (age group, gender, place of residence) using the chi-square test. The different ratings of the four photos in relation to the core values, sympathy, competence, trustworthiness, practice choice, and teamwork being the dependent variables, and the four age/gender subgroups were tested for statistical significance (*p* < 0.050) using multivariate analyses of variance (MANOVA). Here, the main effect of MANOVA was considered, while the difference between each of the four age/gender subgroups was calculated post hoc using the Bonferroni correction. All data analyses were performed using IBM SPSS Statistics (version 28).

## 3. Results

### 3.1. Sample

The sample of 906 participants was composed of 642 women (70.86%) and 264 men (29.14%) who differed significantly (Chi^2^ = 51.32, df = 4, <0.001) in their assignment to the five age groups surveyed (18–27, 28–37, 38–47, 48–57, and >57 years). To control for and clarify this sampling effect, four subgroups were formed with women and men aged 18–47 years and women and men over 47 years (Chi^2^ = 32.59, df = 1, <0.001). The distribution of frequencies can be seen in Figure 2, with more than two-thirds of the women younger than 48 years (*n* = 446, 69.47%) and 196 (30.53%) older than 48 years. The men are almost equally distributed between the two age groups with 131 (49.62%) under and 133 (50.38%) over 48 years. 

### 3.2. Sympathy

When assessing the four photos according to sympathy with different clothing of the practice team, age and gender effects became apparent. The visualization of Table 1 can be seen in Figure 3. Regardless of clothing, older men >48 years rated the photos with a tunic, coat, and mask as more likable than younger men aged 18–48 years, who in turn rated them more likable than older women >48 years. Younger women rated these photos with the lowest mean score for sympathy (see Table 1). Single-factor MANOVA showed a statistically significant difference with a large effect size between the four age/gender subgroups for likability (F = 6.20, *p* < 0.001, η^2^ = 0.028, Wilk’s Λ = 0.943). Post hoc tests (Bonferroni) revealed that the mean differences between the four subgroups were significant (all *p* < 0.050), except for the comparison between younger men and older women.

### 3.3. Competence

Age and gender effects were also evident when the four photos were assessed for competence with different clothing worn by the practice team. The content of Table 2 is visualized in Figure 4. Regardless of clothing, older men >48 years rated the photos with a tunic, coat, and mask with higher competence than younger men aged 18–48 years, who in turn rated them with higher competence than older women >48 years. Younger women rated these photos with the lowest mean score for competence (see Table 2). Single-factor MANOVA revealed a statistically significant difference, with a small effect size between the four age/gender subgroups for combined liking (F = 2.55, *p* = 0.007, η^2^ = 0.008, Wilk’s Λ = 0.975). Post hoc tests (Bonferroni) revealed that mean differences between younger women and the other three subgroups were significant, as was the comparison between older men >48 years and older women (all *p* < 0.050). Younger and older men did not differ significantly.

### 3.4. Trust

When assessing the four photos for trust with different clothing of the practice team, age and gender effects again become apparent. The contents of Table 3 can be seen in Figure 5. Regardless of clothing, older men >48 years rated the photos with a tunic, coat, and mask more trustworthy than younger men aged 18–48 years, who in turn rated them more trustworthy than older women >48 years. Younger women rated these photos with the lowest mean trust score (see Table 3). Single-factor MANOVA showed a statistically significant difference with large effect size between the four age/gender subgroups for combined liking (F = 5.22, *p* < 0.001, η^2^ = 0.017, Wilk’s Λ = 0.950). Post hoc tests (Bonferroni) revealed that mean differences between younger women 18–47 years and the other three subgroups were significant (all *p* < 0.050), as was the comparison between older men >48 years and older women >48 years. Younger and older men did not differ significantly.

### 3.5. Choosing a Practice

When evaluating the four photos for choosing a practice with different clothing of the practice team, age and gender effects again become apparent. The visualization in Figure 6 refers to the contents of Table 4. Regardless of clothing, older men >48 years rated the photos with a tunic, coat, and mask more favorably than younger men aged 18–48 years, who in turn rated them more favorably than older women >48 years. Younger women rated these photos with the lowest mean value for seeking out a practice (see Table 4). Single-factor MANOVA showed a statistically significant difference with large effect size between the four age/gender subgroups for combined liking (F = 5.96, *p* < 0.001, η^2^ = 0.019, Wilk’s Λ = 0.943). Post hoc tests (Bonferroni) revealed that mean differences between younger women and both subgroups of older men and older women were significant, as was the comparison between older men and older women (all *p* < 0.050). Younger women and younger men only tended to differ (*p* = 0.059), and younger and older men did not differ significantly.

### 3.6. Team Participation

When assessing the four photos for team participation with different clothing of the practice team, age and gender effects become apparent. Table 5 is visualized in Figure 7. Regardless of clothing, older men >48 years rated the photos with a tunic, coat, and mask more favorably than younger men aged 18–48 years, who in turn rated them more favorably than older women >48 years. Younger women rated these photos with the lowest mean value for team participation (see Table 5). Single factor MANOVA showed a statistically significant difference with large effect size between the four age/gender subgroups for combined liking (F = 5.65 *p* < 0.001, η^2^ = 0.018, Wilk’s Λ = 0.946). Post hoc tests (Bonferroni) revealed that only the mean differences between younger women and younger men were significant.

### 3.7. Residence

There are also clear effects of age and gender on the choice of place of residence. A total of 440 (48.57%) female participants lived in a more urban setting, and 202 (22.3%) lived in a more rural setting. In total, 212 (23.4%) men lived in an urban setting and 52 (5.73%) in a rural setting (Chi^2^ = 14.00, df = 3, <0.002). The distribution among age groups is documented in Table 6.

Only in the evaluation of the four photos for teamwork with different clothing of the practice team do the effects of place of residence become clear. The contents of Table 7 are shown in Figure 8. Regardless of clothing, more urban residents rated the photos with a tunic, coat, and mask more favorably than residents that are more rural. The mean rating of the photo with pink socks did not differentiate between the two subgroups (see Table 7). The single factor MANOVA showed a statistically significant difference with a small effect size between the four age/gender subgroups for combined liking (F = 2.87 *p* = 0.035, η^2^ = 0.018, Wilk’s Λ = 0.009).

### 3.8. Summary of the Results

Significant influences of the different clothing of interprofessional teams on the perception of sympathy, competence and trust, elective practice, and desired team are found, partly, however, with a small effect due to the size of the sample. In interpreting the results, we must consider the large number of female participants.

Even though our results are conclusive and partly congruent to the results of previous studies, we were able to show in the subgroup analyses that many effects/impacts of different clothing must be reinterpreted in the context of the vast changes taking place right now in our society. In general, the white coat expresses professionalism and competence. The photo with the “pink socks“ mostly symbolizes sympathy and cooperation. It is highly exciting that the wearing of masks generates a feeling of competence and trust across all age groups. However, men seem to have a greater mask problem than women. Young women seem to rely more on sympathy than competence. They are more likely to consider which team to use. As a conclusion, the aspect of team culture seems to be the most important point for young women. They seek care, a common path. They believe more strongly in the community and are not as afraid as the other age groups. Men, on the other hand, without age group preference, look for competence and trust in crises. This is where security is generated for them. This fact also applies overall to the older age groups of both sexes. Regardless of clothing, more urban residents rated the photos with a tunic, white coat, and mask more favorably than residents that come from a more rural area.

## 4. Discussion

Is there one right dress for the physician or for the interprofessional team in a family practice, and how is attire significantly perceived? Do we have to rewrite the story of doctors’ attire and the dominance of the white coat after for the first time having looked at a practice team’s clothing? Only three of the studies conducted in this field were from mainland Europe. Two were from the United Kingdom. Of the more than 50 articles reviewed, few dealt with patients in an ambulatory care setting, most comparable to the patient population in a German family practice [23,24,25,26]. No German or international study had yet examined the effect of how the varying attire of an interprofessional team was perceived. Because of these facts, we thought it to be time to initiate further research with respect to team member attire in a preclinical ambulatory setting in a German practice.

Studies over the past decades have attempted to answer this question, portraying single doctors in their respective attire. One of the largest studies on this topic meta-analyzed 30 studies from 14 countries with over 11,000 patients who met the inclusion criteria [27]. As a result, there was a large dependence on geographic location, patient age, and type of medical care. A majority of patients appeared to prefer traditional physician attire. In the largest study to date to understand patient preference regarding physician attire with over 4000 patients in 2017, it was stated that patients had very high expectations of physician attire, depending on the context of care and geographic location in the country [8]. In a 2004 study, being able to identify the physician by a white coat was important to over 54% of patients over the age of 70 [28]. Our results here are mostly congruent with the above studies mentioned. In our age subgroup analysis, men over the age of 48 rely most on team members wearing a white coat to generate sympathy, competence, and trustworthiness. They were followed by young men, older women, and young women. Wearing a white coat or a mask is the least important for young women.

In total, 43.6% of respondents indicated that how “happy” they were with their treatment depended on their physicians’ attire [26]. This fact also holds true for the photo of our team members wearing pink socks. Here, sympathy and wanting to be part of the team is the main aspect for all age/gender subgroups. Interestingly for young women, trust is also generated through this picture.

While in the period from 1987 to 2015, most studies showed a clear advantage for the classic physician attire, with the dominance of the classic white physician coat, the picture has changed in recent years. There seems to be good evidence for the fact that it is important to specifically adapt healthcare professionals’ attire to the environment of care and the associated expectations of patients [29,30,31,32]. In 2021, an interesting study from the USA showed for the first time the connection between gender, clothing, and role identification [33]. In our study, young women seemed to rely more on sympathy than competence. This points to a completely new aspect of team/physician attire. Young women are more likely to consider which team to use. As a conclusion, the aspect of team culture seems to be the most important point for young women. They believe more strongly in the community and are not as afraid as the other age groups, especially older men.

A survey of advanced-stage tumor patients was able to rule out a relationship between physician professionalism and compassion and professional attire [34]. Likewise, physician attire did not play a role in the perceptions of patients in a military eye clinic [10]. In the setting of a Belgian family physician study, the white doctor/physician coat was perceived as an additional barrier in the doctor–patient relationship. A study in the setting of three U.S. Family care practices failed to see a consistent patient preference for the attire of their attending physicians in over 432 patients* [24]. In addition, a big smile made all the difference regarding physician attire in a study from New Zealand, where patients preferred less formal physician attire [35]. In our study, a less formal aspect of team attire made a positive difference for people living in a more urban area, possible reasons being socio-cultural related.

However, during the coronavirus pandemic, patients accepted scrubs combined with an FFP2 mask as part of personal protective equipment [18]. Through our study, we gained a deeper knowledge of this aspect by looking into the subgroups of our respondents. In general, men seem to have a greater mask problem than women. On the other hand, we found that men without age preference looked for competence and trust in crises. These values were symbolized by a white coat and a mask generating security for them. Overall, the security aspect holds true for the older age groups of both sexes. As far as the self-image of the medical profession is concerned, white coats still seem to have a high symbolic value. This is demonstrated by the relatively new but numerically increasing ritual of the “White Coat Ceremony,” in which medical students and physicians in residency are donned with a white coat in a ceremonial university ritual [36,37].

## 5. Limitations and Future Research Directions

A limitation of our study is the high percentage of women versus the low percentage of men. The distribution of the questionnaire was performed through limited channels only. There was only a limited scenario portraying the team. The depicted attire is typical for clothing in a German context. Business suits, slacks or ties that doctors with international background wear, are not shown. Regarding diversity, only one team member had a different cultural background. Because of the high data output, we did not take the respondents’ education and employment status into account. Neither did we focus on the assignment of team members to their supposed team positions.

A lot of research has been undertaken in the field of single-physician attire. Most of the studies were performed in a clinical setting in the United States or in countries with affiliation to the British medical system. It would be interesting to generate more data from European countries in a more outpatient setting. Working in interprofessional teams is the future of patient treatment. Further studies should concentrate on team attire.

## 6. Conclusions

We asked ourselves if the white coat had become obsolete in a time of great and rapid social change due to the coronavirus pandemic and digitalization, and if there perhaps was a perfect dress for an interprofessional team, which is perceived by the patients as sympathetic, trustworthy, and competent, and at the same time symbolizes that one would like to select the team as a family practice, or even work in it. Our study clearly showed that we should present ourselves visually differently regarding the perception of the patients. For example, if we want to appear medically competent on a practice homepage, we should choose the classic clothing with a white coat. If it is about the presentation in social networks, we should present a picture in a relaxed mood. In medical portals on the internet, we should emphasize the perception of competence and trust. When addressing young female patients, the team aspect and the sense of community are visually important. When addressing men, the traditional dress with doctors in white coats generates safety and competence. Moreover, in the urban environment, it may be the tunic which is perceived much more positively here. Remarkable was the fact that the team members in the photo with classical clothing, white coat, and mask were perceived as significantly more competent. This finding must certainly be interpreted in the context of the persistent coronavirus pandemic. Thus, the masks worn here convey more security for the patients, which in our view clearly contributes to the perception of the increased professional competence of the team.

## Figures and Tables

**Figure 1 healthcare-10-02109-f001:**
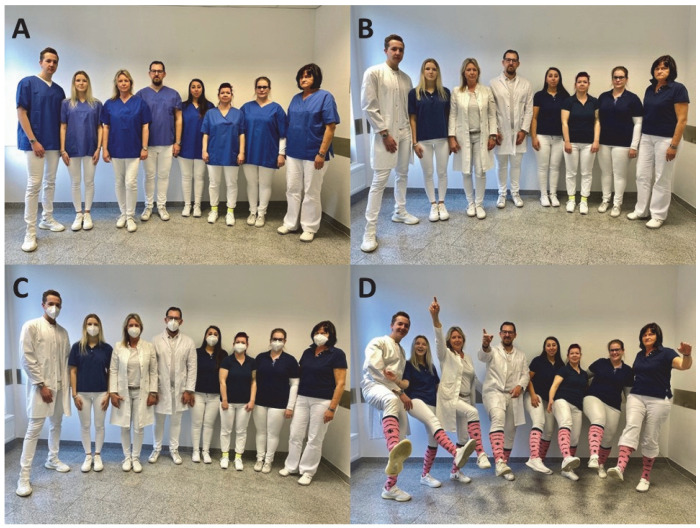
(**A**) All members of the practice team in white trousers with a blue tunic as a top. (**B**) Team members either in white trousers, white polo shirt, and white coat, or in whiter trousers and blue polo shirt (**C**) Team as in the second photo, but with additional FFP2 protective mask. (**D**) Team in a relaxed mood with #pinksocks as an additional accessory.

**Figure 2 healthcare-10-02109-f002:**
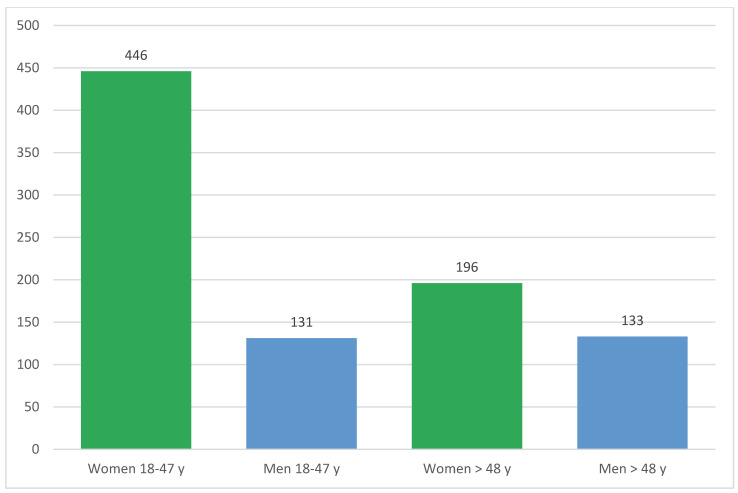
Distribution of the sample according to age groups and gender (*n* = 906).

**Figure 3 healthcare-10-02109-f003:**
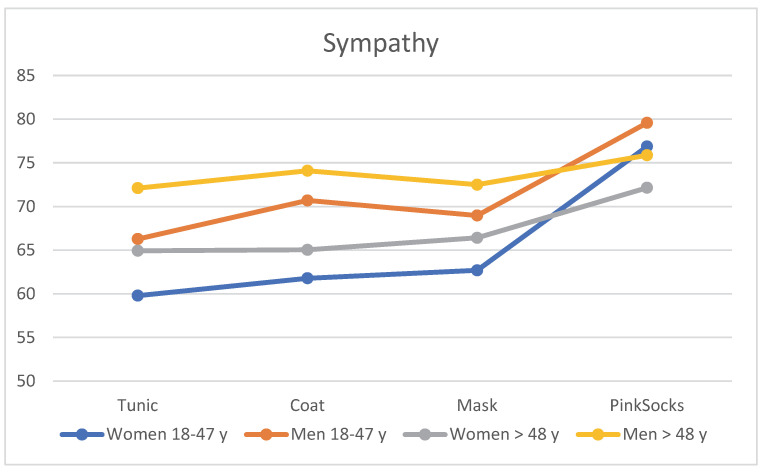
Mean rating of the four photos for likability by the four subgroups (*n* = 906).

**Figure 4 healthcare-10-02109-f004:**
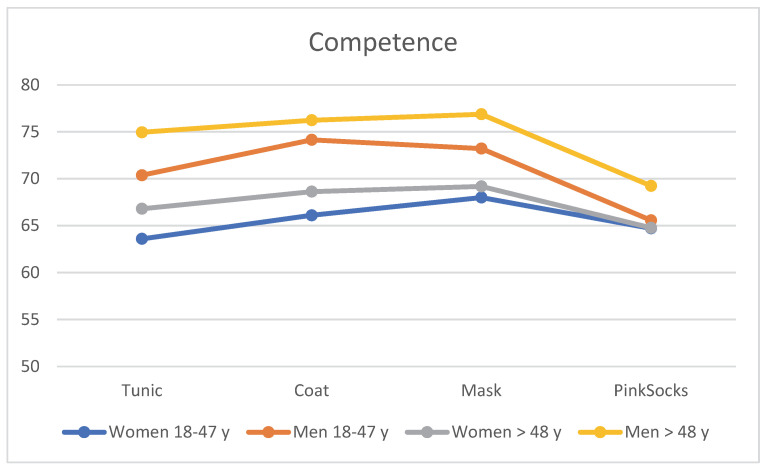
Mean rating of the four photos for competence by the four subgroups (*n* = 906).

**Figure 5 healthcare-10-02109-f005:**
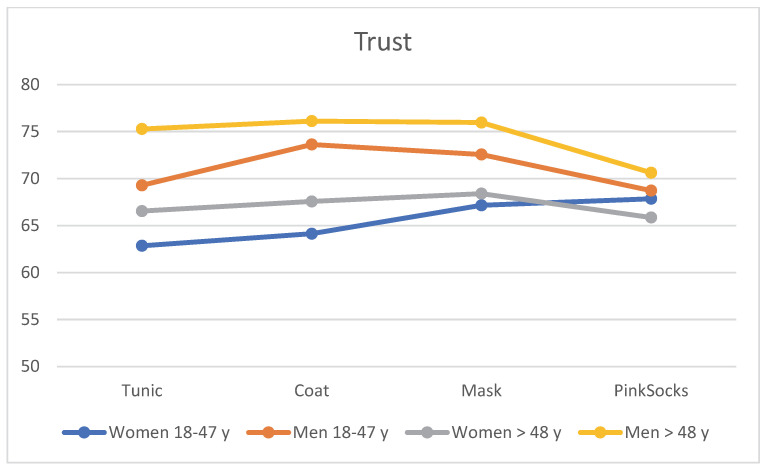
Mean rating of the four photos for trust by the four subgroups (*n* = 906).

**Figure 6 healthcare-10-02109-f006:**
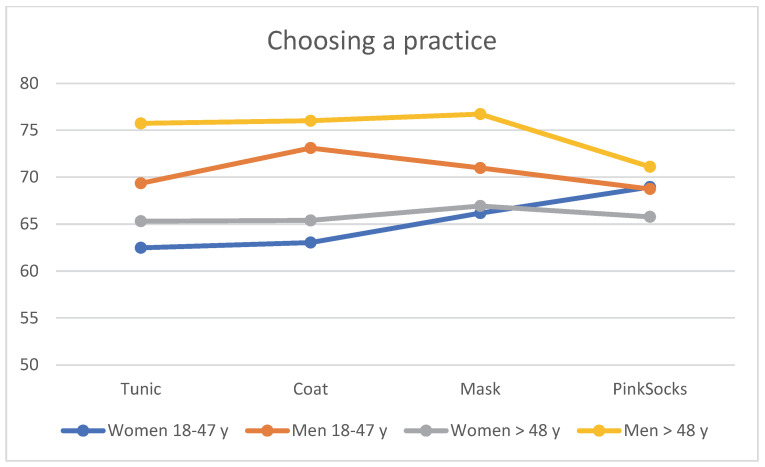
Mean rating of the four photos for choosing a practice by the four subgroups (*n* = 906).

**Figure 7 healthcare-10-02109-f007:**
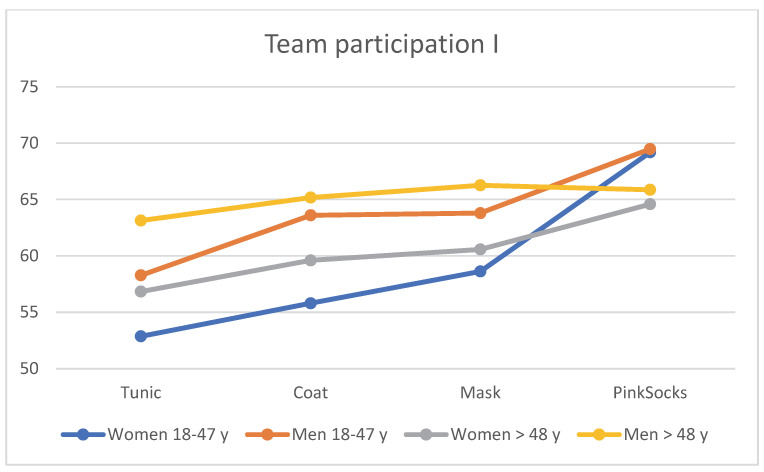
Mean rating of the four photos for team participation by the four subgroups (*n* = 906).

**Figure 8 healthcare-10-02109-f008:**
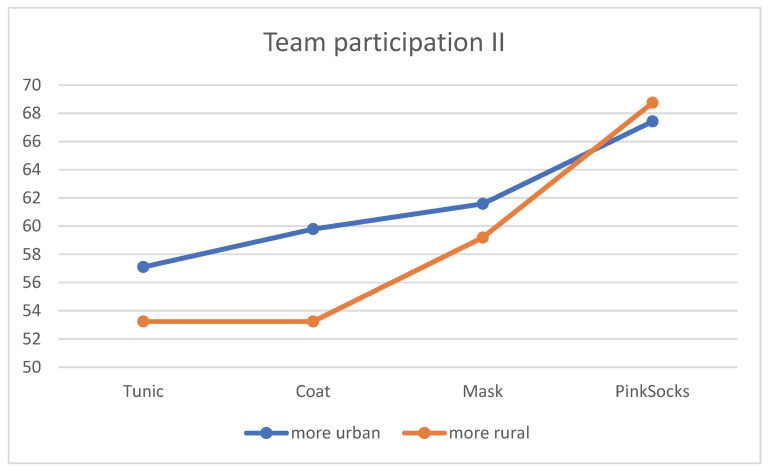
Mean rating of the four photos for team participation by the two subgroups according to place of residence (*n* = 906).

**Table 1 healthcare-10-02109-t001:** Evaluation of the four photos according to sympathy (mean, standard deviation, *n* = 906).

	Tunic M ± SD	Coat M ± SD	Mask M ± SD	PinkSocks M ± SD
Women 18–47 y	59.79 ± 21.99	61.78 ± 20.18	62.69 ± 20.93	76.87 ± 21.01
Men 18–47 y	66.28 ± 24.82	70.69 ± 21.56	68.97 ± 20.51	79.58 ± 22.51
Women >48 y	64.94 ± 25.24	65.04 ± 24.53	66.41 ± 23.49	72.15 ± 25.53
Men >48 y	72.11 ± 26.12	74.09 ± 23.01	72.50 ± 24.05	75.87 ± 27.40
Total	63.65 ± 24.13	65.58 ± 22.26	65.84 ± 22.18	76.10 ± 23.35

**Table 2 healthcare-10-02109-t002:** Evaluation of the four photos according to competence (mean, standard deviation).

	Tunic M ± SD	Coat M ± SD	Mask M ± SD	PinkSocks M ± SD
Women 18–47 y	63.59 ± 19.45	66.10 ± 18.63	68.00 ± 18.98	64.71 ± 20.75
Men 18–47 y	70.37 ± 20.76	74.15 ± 19.16	73.21 ± 21.08	65.56 ± 26.11
Women >48 y	66.80 ± 23.08	68.62 ± 22.27	69.18 ± 22.44	64.77 ± 25.16
Men >48 y	74.95 ± 22.95	76.23 ± 21.94	76.87 ± 21.22	69.24 ± 29.27
Total	66.93 ± 21.36	69.29 ± 20.40	70.31 ± 20.63	65.51 ± 23.95

**Table 3 healthcare-10-02109-t003:** Evaluation of the four photos according to trust (mean, standard deviation).

	Tunic M ± SD	Coat M ± SD	Mask M ± SD	PinkSocks M ± SD
Women 18–47 y	62.85 ± 19.59	64.13 ± 19.56	67.15 ± 19.99	67.85 ± 21.15
Men 18–47 y	69.28 ± 21.38	73.62 ± 20.10	72.56 ± 20.25	68.71 ± 24.87
Women >48 y	66.54 ± 23.64	67.56 ± 24.00	68.40 ± 23.26	65.86 ± 25.24
Men >48 y	75.27 ± 24.00	76.11 ± 23.41	75.96 ± 22.92	70.62 ± 28.46
Total	66.40 ± 21.86	68.00 ± 21.72	69.49 ± 21.43	67.95 ± 23.80

**Table 4 healthcare-10-02109-t004:** Evaluation of the four photos according to choosing a practice (mean, standard deviation).

	Tunic M ± SD	Coat M ± SD	Mask M ± SD	PinkSocks M ± SD
Women 18–47 y	62.47 ± 22.90	63.06 ± 21.53	66.17 ± 29.94	68.96 ± 23.40
Men 18–47 y	69.35 ± 24.65	73.11 ± 21.19	70.98 ± 21.44	68.74 ± 26.45
Women >48 y	65.31 ± 27.86	65.39 ± 26.70	66.93 ± 26.78	65.77 ± 27.38
Men >48 y	75.73 ± 25.63	76.01 ± 23.89	76.73 ± 23.01	70.11 ± 29.59
Total	66.03 ± 25.10	66.92 ± 23.55	68.58 ± 22.97	68.40 ± 25.71

**Table 5 healthcare-10-02109-t005:** Evaluation of the four photos according to team participation, age and gender (mean, standard deviation).

	Tunic M ± SD	Coat M ± SD	Mask M ± SD	PinkSocks M ± SD
Women 18–47 y	52.87 ± 23.12	55.80 ± 22.24	58.62 ± 22.09	69.19 ± 24.41
Men 18–47 y	58.28 ± 24.42	63.60 ± 23.69	63.79 ± 21.53	69.48 ± 26.51
Women >48 y	56.84 ± 26.85	59.60 ± 26.65	60.57 ± 26.62	64.59 ± 29.14
Men >48 y	63.14 ± 29.16	65.17 ± 27.96	66.26 ± 27.38	65.86 ± 31.49
Total	56.02 ± 25.32	59.13 ± 24.59	60.91 ± 24.01	67.75 ± 26.94

**Table 6 healthcare-10-02109-t006:** Distribution of the sample according to age groups and gender by place of residence (number, percent ^(1)^).

	More Urban #%	More Rural #%	Total #%
Women 18–47 y	309 (34.11%)	137 (15.12%)	446 (49.23%)
Men 18–47 y	109 (12.03%)	22 (2.43%)	131 (14.46%)
Women >48 y	131 (14.46%)	65 (7.17%)	196 (21.63%)
Men >48 y	103 (11.36%)	30 (3.31%)	133 (14.68%)
Total	652 (71.96%)	254 (28.06%)	906 (100%)

^(1)^ All figures in percent refer to the sample (*n* = 906).

**Table 7 healthcare-10-02109-t007:** Evaluation of the four photos by the two subgroups according to team participation and place of residence (mean, standard deviation).

	Tunic M ± SD	Coat M ± SD	Mask M ± SD	PinkSocks M ± SD
More urban	57.10 ± 25.57	59.79 ± 24.84	61.58 ± 24.14	67.43 ± 27.59
More rural	53.24 ± 24.51	57.42 ± 23.89	59.19 ± 23.65	68.57 ± 25.22
Total	56.02 ± 25.32	59.13 ± 24.59	60.91 ± 24.01	67.75 ± 29.94

## Data Availability

The data presented in this study can be obtained on request from the corresponding author. At present data are not openly available due to oncoming analyses for further research questions.

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
