# Peer review of "Clothes Make the Man—What Impact Does the Dress of Interprofessional Teams Have on Patients?"

_healthcare, 2022, doi:10.3390/healthcare10102109_

Round 1
Reviewer 1 Report
The paper focuses on an interesting topic, analyzing the impact of physicians' clothing on the doctor-patient relationship. In detail, the paper aims to demonstrate whether physicians' clothing influences patient perceptions with reference to their empathy, competence, and trust.
This topic has already been addressed in the literature. In this article, the authors make a contribution to the literature by introducing the current scenario of the COVID-19 pandemic as a new element. However, it would be appropriate to add the literature section, the research question, highlighting the research gap to be filled.
The literature review section is essential for understanding the studies done on the topic up to date and the future research directions proposed by scholars of the subject.
The methodology adopted is online survey methodology. No detailed information is provided on the survey structure. For example, the authors state that they administered a survey consisting of 52 questions, and among them, two refer to the pandemic. However, it is unclear how the questions were structured and the framework used. It would be helpful to understand, for each of the questions asked, what information they aim to obtain. Since these are as many as 52 questions, it is unclear how the authors analyzed the data obtained. An appropriate criteria for comparing and relating the information collected should be identified. In addition, the authors do not describe the limitations and advantages of the survey methodology, making no reference to the literature describing this method of investigation.
The results and discussion section is well organized. However, it is recommended that the following sections be introduced:
- Conclusions
- Practical and theoretical implications
- Limitations and future research directions
Author Response
Dear reviewer,
thank you for reviewing my paper and supporting me in terms of giving me valuable advice. I thoroughly revised the paper regarding your comments. Please kindly note, that the paper is focused on "team attire" not on physicians clothing. Kind regards H.J. Roehrens
Reviewer 2 Report
This study examines the impact of the dress of interprofessional teams on patients through a questionnaire distributed in German and English. The research is interesting, well-organized, and readily. I have some comments need to be addressed as follows:
Major comments:
Introduction:
line39-41, I feel confused about this sentence, do the authors want to express the data from European research on this topic is sparse and limited and only two questionnaires have been done in Swiss and Belgian? And what are their findings? If related to this topic, they should be described in the introduction.
Some paragraphs only have 1-2 sentences, they may need to be reorganized to less paragraphs.
The introduction needs to cover/expand the related work thoroughly, including the related findings, how this topic is different and readily, and what kinds of factors have been choose or not chosen, and related findings.
Methods:
Figure 1B, there are three doctors with white top.
Line66, I didn’t see the appendix. Here may need a brief description about how the questions look like so that readers have some basic idea about the design of the questionnaire.
line86-87, Where is the link of the questionnaire specifically sent to? Saying the mail distribution lists and social media is not enough as this may affect results significantly.
Line 96, be specific about what kinds of multivariate analysis of variance are used in this study.
Line 69-70, the paper described the sociodemographic data including educational qualification and professional field of activity were collected, however, only the age and gender were tested, and the results were reported. Clarify why only these two are chosen and describe them in the corresponding sections.
Discussion:
Part of the descriptions from previous studies could be moved to the introduction part. This part should focus more on the discussion of the results reported.
The paper has some limitations, for example, the data sample (high percentage of women vs low percentage of men), the way to distribute questionnaire, limited scenarios of practice team, and what kinds of practice.
Minor revisions:
Descriptions of figures in the main text should be consistent, for example, line 73 uses Fig. 1, line 105 uses Figure 5.
line54, what does the “*” refer to? So as the others?
line105, it should be Figure 2 rather than 5.
Figure 3 is a visualization of the results reported in Table 1, but there is no description or even mention of the figure in the text. Figure 4-8 is the same.
Author Response
Dear reviewer,
thank you for reviewing my paper and supporting me in terms of giving me valuable advice. I appreciated your detailed comments, which effectively helped me revising the paper. The "*" was a remnant of the German original with its gender form for male/female patients. Kind regards H.J. Roehrens

Reviewer 3 Report
This research aims to better understand how physician attire is perceived by men and women via quantitative survey responses to 4 pictures of a family practice team showing some team members in different types of attire.
Strengths: The study used an online survey method and had 906 evaluable responses which gives a fairly large sample size. The investigators gave a reasonable rationale for why they combined the age groups of men and women. Statistical analyses are appropriate.
Weaknesses: The paper is not well written and is very difficult to understand in many sections. The methods do not clearly explain who the survey was distributed to and how the distribution lists were determined. The choice of photographs is problematic. Physicians, as described in the manuscript, are mostly male. The selections of attire are not all-encompassing of what might be seen in practice (slacks, shirt and tie or business attire). In the appendix the survey has the respondent assign profession to each person in photos 1 and 2. However, the data are presented as if physicians were already designated in the survey. See legend to Figure 1B, where it says that doctors are in white trousers but the appendix is asking the respondent to choose what profession each person is. The rigor of this study is lacking and is not acceptable for publication in Healthcare.
Author Response
Dear reviewer,
thank you for reviewing my paper and supporting me in terms of giving me valuable advice. I thoroughly revised the paper regarding your comments. Please kindly note, that the paper is focused on "team attire" not on physicians clothing. Kindly let me you inform you, that it is not common to wear business suits, ties or shirts in German medical practices. I newly commented on that fact in the paper. Regarding the photos the respondents had to assign a supposed position to each person. Persons wearing white coats were not identified as doctors. I revised the subtitle in photo B. The respondents did not know, if the persons dressed in white coats were really doctors. I further concentrated on the importance and perception of wearing a FFP mask. No research on the field of team attire has ever bee done before! In my opinion these facts give the research paper a very good rigor and is worth being published in Healthcare.Kind regards H.J. Roehrens
Reviewer 4 Report
Thank you for the opportunity to review your paper. It is an interesting work which can be enriched by more scientific support.
1. The introduction should be revised to establish a clearer and more compelling motivation for the study.
2. More specific and realistic (substantial) implications are required. It is difficult to recognize difference from the already-preceded research. In addition, even the supplementation is necessary for research limitations and future research.
Author Response
Dear reviewer,
thank you for reviewing my paper and supporting me in terms of giving me valuable advice. I thoroughly revised the paper regarding your comments. The introduction is now clearer and more compelling. The section of limitations and future research was added. More realistic implications were implemented. The discussion section was re-structured. Kind regards H.J. Roehrens
Round 2
Reviewer 1 Report
Although the authors expanded the literature on the topic and highlighted the contribution offered (the effects of the FFP2 mask on patients), the literature review section is missing. The authors should include a section, after the introduction, on the analysis of the literature on the topic. Instead, the introduction should be set up as follow: what the problem is and why it is interesting, compared to the relevant literature; introduction of the objective of the work, the methodology adopted, indication of the expected results and implications in terms of contribution offered to the existing literature; and finally, indicate a road map, i.e., the summary of the structure of the article. In addition, in the introduction section there must be a link to the papers analyzed in the literature review section. Also in this version of the paper, the research methodology section does not illustrate the description of the limitations and advantages of the survey methodology, making no reference to the literature describing this research method. The practical and theoretical implications should be added in the conclusion.Author Response
Dear Reviewer,
thank you again for your reviewing our paper.
Let me please point out, that not only did we focus on FFP2 masks but mainly on the topic of an interprofessional team and how it was perceived. This topic had never been investigated in literature before. This being the reason why in our introduction we first looked into the existing literature which deals with single physicians attire. All major studies and their outcome were subsequently introduced and analysed. The introduction of the objective was also properly introduced. ("Although a great body of research....handprotection"). In this passage further explanation of the methodology was given through this second revision.("Therefore , we designed our questionnaire..subgroups subsequently"). Discussion of our expectation was added. The limitations and advantages of the survey methodology is discussed in the "method" section. Literatur is being referred to. We think that the paper is now well balanced and ready for publication. This is the feedback we also got from the other reviewers.
Kind regards H.J. Roehrens
Reviewer 2 Report
The authors did a good job to address most of my comments and I only have a few minors listed below.
Minor comments:
Put three paragraphs from line 100 to 110 to one paragraph as the second and third only have 1-2 sentences.
Line 142, put abbreviation of MANOVA after “multivariate analyses of variance”.
For my question why authors only chose age and gender for analysis in Line 69-70 in the first round, I don’t see the appropriate response of this from Line 71-80. But the limitation section mentions it because of the high data output of the questionnaires. Perhaps add 1-2 sentences in the introduction to clarity it.
There are several texts using “*”, what does it refer to? I don't see the explanation.
Author Response
Dear reviewer,
thank you again for reviewing our paper and commenting on it.
It seems that your line coding was different from ours, but we think we revised all your suggestions. The three paragraphs were combined into one. "MANOVA" was added. We also introduced two lines in the introduction section to clarify the study design. ("Therefore, we designed our questionnaire..."). There should be no more "*" in the script. In the German original draft they were used in combination with the word "patients" as so-called gender stars referring to gender correct language.
Kind regards
H.J. Roehrens
Reviewer 3 Report
The authors have significantly improved the manuscript.
Strengths include more detail on how the photos were presented in the survey, , how the survey was deployed, correction of the numerous errors in the legends and text, and more clarity and accuracy in interpretation of results. The limitations section is well done.
Weaknesses have been addressed in the limitations section and help the reader put the results into context.
Author Response
Dear reviewer,
thank you again for reviewing our paper and commenting on it.
Kind regards
H.J. Roehrens